# Multiple Myeloma and Thrombosis: Prophylaxis and Risk Prediction Tools

**DOI:** 10.3390/cancers12010191

**Published:** 2020-01-13

**Authors:** Despina Fotiou, Maria Gavriatopoulou, Evangelos Terpos

**Affiliations:** Department of Clinical Therapeutics, National and Kapodistrian University of Athens, School of Medicine, 11528 Athens, Greece; desfotiou@med.uoa.gr (D.F.); mariagabria@gmail.com (M.G.)

**Keywords:** multiple myeloma, venous thromboembolism, risk assessment models, thromboprophylaxis, direct oral anticoagulants

## Abstract

Thromboembolism in multiple myeloma (MM) patients remains a common complication that renders the optimization of our thromboprophylaxis practice necessary. This review aims to make clear the need for the development of more accurate risk assessment tools and means of thrombosis prevention. Current clinical practice is guided by available guidelines published by the IMWG in 2014, but the extent to which these are implemented is unclear. Recently, several groups developed clinical scores for thrombosis risk in MM in an attempt to improve risk stratification, but these have not been validated or used in clinical practice so far. Research in this field is increasingly focusing on understanding the unique coagulation profile of the MM patient, and data on potential biomarkers that accurately reflect hypercoagulability is emerging. Finally, promising evidence on the effectiveness of direct oral anticoagulants (DOACs) in the context of thrombosis prevention in MM patients is increasingly becoming available. The critical appraisal of the above research areas will establish the necessity of combining disease-specific clinical risk factors with coagulation biomarkers to allow more effective risk stratification that will eventually lead to the reduction of this significant complication. Results from ongoing clinical trials on the role of DOACs are much anticipated.

## 1. Introduction

The extraordinary advances in the therapeutic armamentarium available for patients with a new diagnosis of multiple myeloma or relapsed/recurrent disease has led to significant increases in overall survival (OS) but has also drawn attention to the management of treatment-related complications for these patients. Among the commonest complications seen in this population is venous thromboembolism (VTE), as more than 10% will develop VTE during the course of their disease [1,2,3,4].

Data from studies that link VTE and inferior overall survival (OS) in MM patients are conflicting, and a clear association has not been established [3,5,6,7]. However, thrombotic events do have an adverse impact, as they may lead to treatment interruption, increased morbidity, and add to the economic burden of the disease in the population [8,9]. There is a lack of studies that have attempted to specifically assess the economic burden associated with VTE occurrence in MM patients. Data from other cancer patients demonstrate as expected increased costs associated with the long-term use of pharmaceutical agents for the treatment of thrombosis, the need for hospitalization, and increased risk of complications as well as adverse effects on patient’s quality of life [8]. Given the significant improvement in the OS of MM in the era of novel agents, the conversation regarding the price and affordability of current treatments is becoming increasingly available. Formal pharmacoeconomic analyses are required to assess the cost-effectiveness of treatment options and the financial burden of managing the complications and adverse effects of these therapeutic agents, including the management of VTE [10].

Thrombogenicity in MM is multifactorial, and risk factors are traditionally distinguished in three groups [11,12]: patient-related clinical risk factors, disease-related risk factors, and treatment-related risk factors. It has become evident from clinical trial data during the last decade that immunomodulatory agents among anti-myeloma treatments stand out as having a considerable prothrombotic effect. Recognizing the significant risk associated with the use of immunomodulatory agents (IMiDs), the International Myeloma Working Group (IMWG) 2014 statement [13], and the European Myeloma Network Guidelines in 2015 [14] both included guidance on the prevention of VTE in MM patients who receive IMiDs. The risk stratification algorithm proposed is based mostly on expert opinion and the available data from clinical trials [15,16,17,18,19,20]. The National Comprehensive Cancer Network (NCCN) guidelines use a similar framework and include patients that receive non-IMiD-based regimens [21] (Table 1). 

These guidelines have been available since 2014; however, data from clinical trials demonstrate that the rates of residual VTE remain high [22,23,24]. Therefore, it is safe to conclude that the current risk stratification is suboptimal and fails to fully capture and distinguish between low, intermediate, and high-risk MM patients for VTE. At the same time, the extent to which the guidelines are implemented in everyday clinical practice can be questioned, increasing the complex task of assessing its effectiveness. Recent publications seem to support that most physicians tend to apply thromboprophylaxis based mostly on clinical experience. In a recent report, the rate of compliance with guidelines was only 66% in a cohort of patients who received lenalidomide-based regimens. 

This review aims to highlight the multifaceted nature, the complexity, and heterogeneity that characterizes the prothrombotic environment that exists in the MM patient. It aims to demonstrate that optimum risk stratification and effective thromboprophylaxis can only be achieved through the development of a myeloma specific risk assessment models (RAM) for VTE. A RAM that includes clinical and treatment-specific risk factors in combination with disease-specific coagulation biomarkers can potentially successfully capture all aspects of the heterogenous prothrombotic environment that exists in MM patients. Research efforts need to further focus on the exploration and understanding of the interplay between markers of plasma and cellular coagulation and the MM microenvironment [25]. Following effective risk stratification, the most effective and safe tool for thromboprophylaxis needs to be established: the right agent for the right patient and for a sufficient amount of time. Increasingly, direct oral anticoagulants (DOACs) are gaining ground in the field of thrombosis treatment and venous thromboembolism prophylaxis. Randomized controlled trials are required that can provide robust data that support their use in the context of VTE prophylaxis in MM.

## 2. Understanding the Complex Procoagulant Profile of the MM Patient

To date, the understanding of the underlying processes that lead to enhanced coagulation in the MM patient has not been delineated. Table 2 summarizes the available data linking patient-related, disease-related, and treatment related risk factors with VTE occurrence. 

### 2.1. Patient-Related Risk Factors

Standard VTE risk factors that are specific to the patient’s characteristics, past medical and surgical history, and current medications are included in the risk assessment. Age, renal impairment, immobility, and frequent hospitalizations due to immunoparesis and immunosuppression are all very relevant to the MM patient [32]. Studies have shown that the incidence of common thrombophilic polymorphisms in MM (factor V Leiden and PTG20210A polymorphism) is similar to that of the general population [27,48].

These are all well-recognized VTE risk factors, and there are no studies specific to MM that demonstrate their association with an increased risk of venous thromboembolism. 

### 2.2. Disease-Specific Risk Factors and the Search for a Biomarker

The mechanisms underlying the prothrombotic environment observed in MM are not understood up to date. A number of plasma and cellular biomarkers of coagulation have been studied by several groups at various timepoints prior to, during, and post-treatment initiation. No group has yet identified a biomarker that accurately reflects prothrombotic risk in these patients and can be combined with clinical factors to enhance risk stratification. 

The diagnosis of MM itself is a risk factor, as a newly diagnosed MM patient is at higher risk of VTE compared to a patient with relapsed or recurrent disease. A hypercoagulable environment in MM is sustained by increased levels of inflammatory cytokines and other factors of coagulation. Fibrinolysis and fibrin polymerization is also disrupted due to interference by the monoclonal component [3,12,36,37,38,39,54]. Platelet dysfunction and increased adhesion have been reported in patients with MM, which may also explain the demonstrated efficacy of aspirin as an agent for thromboprophylaxis in MM patients [38,40,41]. Some reports have also shown Lupus Antibody Coagulant (LAC)-like activity by the monoclonal component and the presence of antibodies against antithrombin and protein C and S or resistance to the activated protein C pathway [35,39,42,43,44]. Microparticles (MP), either tissue factor or platelet derived (TF-MP or PDMP), also seem to contribute to the procoagulant environment and are perhaps linked to VTE occurrence [34,55].

Thrombin generation (TG) is being increasingly studied by many groups who perform measurements at baseline and during treatment as well as explore the association with VTE occurrence. Most groups have reported abnormal TG in multiple parameters of the assay compared to healthy controls [56,57]. Data is variable and difficult to compare across studies as different TG assays have been used as well as different TG trigger concentrations and phospholipid reagents. Crowley et al. compared TG in MM patients, patients with Monoclonal gammopathy of undetermined significance (MGUS), and healthy controls, and found endogenous thrombin potential (ETP) to be lower, the lag time shorter, the peak shorter, and the velocity index higher in MM patients [58]. Legendre et al. in 2018 also found TG to be attenuated compared to healthy controls with prolonged lag time and time to peak with decreased peak and ETP [59]. Ay et al. demonstrated a significant association between thrombin peak concentration and time to Peak concentration (ttPeak) and VTE risk [25], and Leiba et al. found significantly higher ETP and peak thrombin concentration in patients who developed VTE compared to those who did not [45]. Another group recently published data on 71 patients with MM and performed a serial analysis of thrombin generation parameters during the first four cycles of treatment. TG parameters remained unchanged throughout treatment irrespective of treatment regimen, but they were significantly higher before cycles 2 and 3 for patients who received IMiDs. No association was determined between baseline levels of ETP, thrombin peak concentration, or time to peak and VTE [29]. In another study of 106 MM patients, the TG capacity was higher in MM patients both in platelet-poor plasma (PPP) and platelet-rich plasma (PRP). In PRP, TG was significantly higher in patients treated with lenalidomide compared to MM patients who did not receive IMiDs [46]. In a recent publication by our group as part of the ongoing ROADMAP-MM-CAT (PROspective Risk Assessment anD bioMArkers of hypercoagulability for the identification of patients with Multiple Myeloma at risk for Cancer-Associated Thrombosis) study, VTE risk was shown to be associated with longer procoagulant phospholipid-dependent clotting time (PPL-ct)^®^ and lower endogenous thrombin potential (ETP) in patients with newly diagnosed multiple myeloma (NDMM). In this cohort of 144 patients, thrombin generation was unexpectedly attenuated compared to healthy controls [60]. 

The search for a useful biomarker continues through the exploration of the complex coagulation of MM patients. 

### 2.3. Treatment-Related Risk Factors

The effects and the contribution of different anti-myeloma agents on VTE risk are the best understood among different risk factors. The use of IMiDs in the era of novel agents (thalidomide and its derivatives lenalidomide and pomalidomide) has been associated with a rise in VTE occurrence in the MM population. Thalidomide or lenalidomide (Len) monotherapy does not contribute significantly to the baseline VTE risk. It is reported to be around 3%–4% but can increase up to 26% with the addition of high dose dexamethasone or multi-agent chemotherapy or anthracyclines [39,40,48,61,62,63,64]. The rates are also low with Len maintenance post-autologous stem cell transplant (ASCT) without thromboprophylaxis, and one group reported a 6% VTE rate during a median follow up of 45 months [65]. The associated VTE risk persists over time and does not decrease as the duration of exposure increases [66,67]. Data on lenalidomide-associated VTE risk are presented more extensively in Table 3. Fewer data exist on thrombotic risk linked to pomalidomide, which is lower compared to lenalidomide but may reflect the current mandatory use of thromboprophylaxis [68]. Reported VTE rates vary depending on the dose of pomalidomide and range from 3–7% with 4 mg pomalidomide combined with dexamethasone to 0%–6% with 2 mg plus dexamethasone [69,70,71]. Among proteasome inhibitors, the use of bortezomib is associated with very low VTE rates and might even have a protective effect when combined with thrombogenic agents [72]. The data on the potential thrombogenic or thromboprotective effects of the second-generation protasome inhibitor, carfilzomib, is not as clear yet, and more studies are required [73]. Increased VTE risk does not seem to be one of the adverse events linked to elotozumab, daratumumab, or ixazomib among the available approved drugs for MM patients [74,75,76,77,78,79].

The exact mechanisms underlying the IMiD-induced thrombogenic effect are not known. Association studies so far have hypothesized a role for increased vonWillenbrand factor (VWF), factor VIII, and tissue factor (TF), which mediate procoagulant effects on endothelial cells. There is also enhanced platelet activation and aggregation and reports for activated protein C resistance mediated by cytokines [80]. Individual immune response and modulation might affect the effect of thalidomide on platelet activation, as immune modulation may lead to an early clearance of activated platelets [81]. High-dose dexamethasone increases the P-selectin, VWF, and FVIII levels [82], and doxorubicin seems to induce a procoagulant phenotype on endothelial cells and to increase the levels of plasma thrombin that is generated [83]. There is some data to support that lenalidomide use upregulated cathepsin G and increases the levels of endothelial stress markers sch as intercellular adhesion molecule 1 (ICAM), plasminogen activator inhibitor -1 (PAI-1), and vascular endothelial growth factor (VEGF). Higher levels of P-selectin, fibrinogen, and homocysteine following lenalidomide treatment have also been reported. The transient thrombocytopenia observed with the administration of bortezomib, and its anti-thrombotic effect is likely to be exerted via the inhibitory effects on the 26 S proteasome [80,84]. 

## 3. Risk Assessment Tools

### 3.1. Guidelines and Clinical Practice

Table 1 summarizes the risk factors included in the algorithm proposed by IWMG, European myeloma network (EMN), and (National comprehensive cancer network (NCCN) guidelines. As discussed previously, the value of these guidelines is questioned given the residual rate of VTE observed in recent trials, despite the use of thromboprophylaxis. In addition, clinicians tend to rely more on their own clinical experience rather than trust and apply the algorithm. In the Myeloma XI study, despite using thromboprophylaxis according to the IMWG guidelines for a minimum of 3 months with low molecular weight heparin (LMWH) for high-risk patients and aspirin for low-risk patients, the VTE rate was 11.8% and was highest during the first six months following diagnosis. In addition, the mode of thromboprophylaxis patients used was often inconsistent with the initial risk stratification [13,15]. Therefore, the proposed algorithm seems to fail to minimize events and optimally identify patients at high risk for VTE. A retrospective data analysis of the implementation and effectiveness of the IMWG guidelines demonstrated that among the patients that experienced a VTE event, 18% had been stratified as low risk prior to treatment initiation at baseline and 82% had been stratified as high risk. There was no association between the initial risk stratification and the mode of thromboprophylaxis of use. Therefore, it was demonstrated that guideline concordance in terms of either aspirin (ASA) or LMWH was lower than expected [22].

### 3.2. Risk Assessment Models

All clinical trials that currently involve the use of IMiDs either in newly diagnosed or recurrent/relapsed disease recommend thromboprophylaxis based on the IMWG guidelines. However, residual VTE rates clearly point out the suboptimal nature of the current tools. In addition, outside clinical trials, the rates of compliance and consistent use of the algorithms are low. More sensitive risk stratification tools are required that can capture all aspects of the prothrombotic profile observed in the MM patient. 

The importance and the clinical benefit of using risk assessment models (RAM) for thrombosis in cancer patients have become established in recent years since the development of the Khorana risk score in 2008 [85]. The Khorana score cannot be extended to MM patients, as it does not accurately predict VTE in the MM population [86]. A RAM specific for MM that includes treatment-related parameters to adequate reflect thrombotic risk is required. The value of incorporating biomarkers into clinical RAMs has been shown previously as the incorporation of P-selectin and D-dimers into the Vienna prediction score improved the sensitivity and specificity of the original Khorana score for chemotherapy-related VTE risk in patients with solid tumors [87].

Two clinical RAMs were published in 2019 using retrospective data from databases. Sanfilippo et al. published in 2019 the IMPEDE VTE risk clinical score for MM patients based on retrospective data from the Veterans Administration Central Cancer Registry in 4446 MM patients (for a definition of IMPEDE, see Table 3). Weighting was applied to various patient-specific risk factors for patients with MM (Table 3). Three risk groups were identified, and the respective six-month cumulative incidence of VTE following treatment initiation was 3.3% for the low-risk group (scores ≤3), 8.3% for intermediate-risk group (score of 4–7), and 15.2% for the high-risk group (≥8 score). The score was externally validated using the Surveillance, Epidemiology, End Results (SEER)–Medicare database and 4256 MM patients [88]. A second group also developed a clinical RAM for MM patients who receive IMiD-based regimens using the same database to extract data retrospectively; 2397 patients with MM were selected initially using the SEER database, and the data were subsequently validated using the Veterans registry. Five variables were included in the SAVED Score RAM (Surgery, Asian race, VTE history, Eighty years old, Dexamethasone) (see Table 3) [89]. Patients were grouped into either low or high risk using this RAM, and the hazard ratios were reported for high versus low VTE risk were 1.85 (*p* < 0.01) and 1.98 (*p* < 0.01), respectively. The authors argue in favor of the higher discriminative power of the SAVED score compared to the algorithm proposed by the NCCN guidelines. Despite the fact that the two scores were developed and validated in similar settings, there are significant differences. One reason could be linked to the fact that the SAVED score was developed selecting only MM patients receiving IMiDs. The methodological approach followed is also not identical. Finally, each score possibly captures VTE risk in a unique manner; however it has significant overlap with the other score, given the particularly multifaceted nature of thrombosis in MM patients. 

## 4. Thromboprophylaxis: To DOAC or Not To DOAC?

Robust clinical data to support the use of one pharmacological agent over the other in MM patients as thromboprophylaxis are missing. Factors to consider are effectiveness and safety as well as convenience. Essential issues in the MM patient also include renal dosing, cut-offs for use in the context of thrombocytopenias, and frailty associated with the elderly. 

The rationale underlying the use of aspirin as thromboprophylaxis in low-risk MM patients who receive IMiDs lies with the evidence that supports enhanced platelet activation induced by IMiDs and altered platelet function in patients with MM [36,40,62,81]. Most clinicians chose the 100 mg dose, despite the lack of robust data to support it. One of the few RCTs ever designed to address the question of thromboprophylaxis in MM did not demonstrate a significant difference in VTE occurrence when the use of aspirin was compared to enoxaparin in a group of MM patients who received IMiD-based regimens [19]. Another RCT that compared ASA and fixed low-dose warfarin (1.25 mg/day) to LMWH (enoxaparin 40 mg/day) as agents of VTE prevention in 667 NDMM patients who received thalidomide also did not demonstrate a significant difference between the three agents; the rate of VTE was 6.3% in the ASA group, 8.2% in the warfarin group, and 5% in LMWH group [18]. The Myeloma XI study included protocol-based thrombosis risk assessment. Among patients who experienced a VTE, 9.2% were on therapeutic dose of warfarin, 44.1% were on LMWH (prophylactic dose), and 31% were on aspirin. However, given the baseline risk stratification, a direct comparison is not possible [24]. The VTE rate of 10.7% versus 1.4% for patients who received aspirin versus LMWH respectively in a recent retrospective review of over 1126 patients demonstrates the suboptimal protective effect of aspirin as thromboprophylaxis even in low-risk patients and adds controversy to its role [90]. Its use is discouraged during the initial months of treatment initiation when the VTE risk is highest for NDMM patients. It remains an option for later timepoints during disease remission [91,92]. 

Prophylactic LMWH is currently the standard of care based on guidelines by the IMWG, EMN, and NCCN and based on approve indications for use of this drug group. Most clinicians favor LMWH compared to warfarin particularly for patients with cyclical cytopenias, who are at higher bleeding risk. Patient compliance given the parenteral method of administration remains an issue. Two other important disadvantages of LMWH compared to warfarin include cost and the need for renal adjustment. 

Currently, the most favored class of drugs are DOACs. They are inhibitors of clotting factors Xa or IIa, they are administered orally, and they do not require blood monitoring at standard doses. DOACs have been licensed for the treatment of cancer-associated thrombosis, but their role is thromboprophylaxis for these patients remains unclear up to date, as there is not enough robust data yet to support this use [93]. In a retrospective review that assessed the safety and efficacy of DOACs (dabigatran, rivaroxaban, or apixaban) versus warfarin in patients on IMiD-based regimens, there were four non-major bleeds in the DOAC group versus six in the warfarin group [94]. One group compared the VTE event rate prior and post 2014 and the introduction of a policy change in their center to use apixaban 2.5 mg twice daily as routine thromboprophylaxis for patients on IMiDs. Before 2014, the VTE rate was 20.7% in patients on aspirin and 7.4% in patients on LMWH compared to no VTE events after 2014 within six months of treatment initiation [95]. There is an ongoing single-arm phase IV study (NCT02958969) that aims to evaluate prospectively the safety and efficacy of apixaban for primary VTE prevention in MM patients. The primary objective is to assess VTE occurrence within six months in patients who receive IMiD-based therapy [96]. At interim analysis at three months, no VTE events and no major hemorrhage was reported [96]. Pergourie et al. also recently presented data from the use of apixaban as prophylaxis in MM patients on IMiDs. Two events were reported among 140 patients receiving apixaban 2.5 mg twice daily over six months [97]. DOACs are substrates of P-glycoprotein and P450; therefore an important issue to note with their use compared to the other classes of drugs is the potential drug–drug interactions. Fortunately, no anti-myeloma agent (excluding dexamethasone) is known to be a potent inhibitor or inducer of these pathways [98,99,100]. However, an additional issue associated with the oral route of administration is polypharmacy, which is very relevant in these patients. 

Important issues to consider when deciding upon the most suitable mode of pharmacological thromboprophylaxis for the MM patients include age and associated frailty, cyclical platelet counts due to bone marrow infiltration, and the cytotoxic effects of chemotherapy in addition to renal clearance. For patients with GFR <30 mL/min, most clinicians opt for unfractionated heparin and warfarin or LMWH adjusted to anti-Xa levels. Both DOACs and LMWHs are contraindicated in patients with a glomerular filtration (GFR) rate <30 mL/min. Patients with end-stage disease are usually excluded from clinical trials; therefore, there is a paucity of data for this subgroup of patients [101]. The summary of product characteristics of each class of DOAC provides information on renal dosing adjustments [101]. Currently, using unfractionated heparin or LMWH adjusted to anti-Xa levels is considered the most legitimate option for patients with end-stage renal disease. As more safety and efficacy becomes available, DOACs are increasingly being opted for on a case-by-case basis, even for these patients [102]. The patient with thrombocytopenia is another challenge, as clear-cut instructions and thresholds for the use of different agents are absent. Most clinicians would use the empirical cut off of 50,000/mm^3^ for the administration of full LMWH administration and would half the dose for platelet counts between 49,000 and 30,000 mm^3^ [103,104,105]. Based again on clinical experience, DOAC administration is considered safe at platelet counts of >50,000/mm^3^ when the indication is treatment of VTE and at >75–80.000/μL when the indication is prophylaxis [106].

Data from ongoing RCTs are much anticipated. Robust evidence that will demonstrate the effectiveness and safety of DOACs and will guide their use among different MM patient populations in the newly diagnosed and relapsed/refractory setting is required. To establish their use in this field, there is also a need for RCTs specifically designed to compared different modes of thromboprophylaxis in MM patients.

## 5. Conclusions and Recommendations

Existing 2014 IMWG guidelines (and 2016 EMN guidelines) propose baseline risk stratification for MM patients on IMiDs and the use of aspirin for low-risk patients and prophylactic dose LMWH for higher-risk patients. The rate of residual VTE rate reported from recent RCTs remains high, signifying the limited power of this risk stratification tool in accurately reflecting all aspects of the diverse procoagulant environment that exists in MM patients [24,60]. In addition, the extent to which the available algorithm is being applied in every day clinical practice is questionable. There is also a lack of formal recommendations for patients on non-IMiD-containing regimens [13,22,24]. There is the need for optimization of the current tool utilizing a risk assessment model (RAM) that combines disease-specific, patient-specific, and treatment specific risk factors to accurately stratify VTE risk and guide thromboprophylaxis. 

The IMPEDE and the SAVED scores for VTE risk are clinical scores that have been developed retrospectively and therefore retain the advantage of a very large dataset. Weighting of the risk factors included is expected to improve their performance comparative to the current IMWG/NCCN guidelines. They both include only patient-specific risk factors and treatment-related parameters. MM-specific parameters are missing from the RAM, although there is currently no evidence to support a direct link between ISS stage, disease burden, cytogenetics, or any other disease characteristic to VTE occurrence. It should be noted that none of the groups make recommendations for thromboprophylaxis based on the proposed risk stratification. They are both simple and easy to calculate, but prospective validation will be required prior to their incorporation into clinical practice. Currently, no risk assessment tool makes a distinction between NDMM and relapsed and/or refractory MM patients (RRMM) patients. A new MM should perhaps in the future be included in RAMs as an additional risk factor. The question of whether the performance of these RAMs can be improved by the incorporation of a biomarker remains to be answered, but they could both serve as a backbone for the incorporation of additional parameters. 

Given the complexities and heterogeneity of the VTE risk in the MM population, some groups have turned their research efforts toward the identification of a generic coagulation biomarker that can accurately reflect VTE risk and can be incorporated into a clinical RAM to increase its sensitivity. Such a task is demanding, given the complex and heterogeneous coagulation profile of the myeloma patient. Thrombin generation, P-selectin, platelet-derived microparticles, and procoagulant phospholipid clotting time are some of the biomarkers that have been studied. Το date, no such biomarker has been identified [45,59,60]. Low-cost and simple assessment tools that do not require high-level expertise are prerequisites for the selection of a suitable biomarker. The prospective ongoing ROADMAP-MM-CAT is exploring the coagulation profile of the MM patient in the attempt to identify a marker of coagulation that can be incorporated into a clinical and disease-specific RAM.

Exploration of the complex interactions between the MM microenvironment and cellular and plasma coagulability should continue, as the understanding of the underlying mechanisms and interactions will eventually allow risk assessment optimization. At the same time, the effect of current and emerging treatments on the underlying pathways should be studied and understood. The inability to identify so far a generic biomarker to accurately reflect the above processes is perhaps a reflection of the complex and heterogeneous coagulation profile of MM patients, which results from the interaction of multiple factors. 

Current recommendations propose the use of aspirin and LMWH. However, DOACs are becoming increasingly popular. Their profile is favorable, as secondary to safety and efficacy, they are administered orally and do not require routine monitoring. They are currently licensed for use in the treatment of cancer-associated thrombosis [93,107,108]. Ongoing trials will most likely establish their use in VTE prophylaxis in ambulatory cancer patients as well, and the results are much anticipated [107]. More prospective data on their use will of course be required to establish their role in the field of thromboprophylaxis. In addition, the disadvantages of using an oral administration route (diarrhea, vomiting, drug-to-drug interactions) should be taken into account. Recently, updated NCCN guidelines for cancer-associated thrombosis and thromboprophylaxis have included DOACs for the first time. Randomized controlled trials (RCTs) are required and designed to provide head-to-head comparisons of different methods of pharmacological thromboprophylaxis. These should use clear-cut risk stratification criteria to allow the generation of robust data on safety and efficacy. In addition, trials that include MM patients with renal impairment and thrombocytopenia are required to address the unanswered question of which mode of thromboprophylaxis to use for these patients. An update of the IMWG and EMN guidelines regarding MM VTE risk assessment and thromboprophylaxis is very much needed and eagerly anticipated. 

### Potential Algorithm for Risk Stratification of Patients

Figure 1 summarizes an ideal/future algorithm for VTE risk prediction. It uses information from current IMWG and EMN guidelines, data from RCTs, emerging data on DOACs, retrospective MM VTE risk prediction clinical scores, clinical experience, and anticipated future advances in the field. The weighting of clinical and disease-specific risk factors should be made based on the IMPEDE risk score, which should be incorporated in the RAM following prospective validation. A biomarker of coagulation that accurately reflects the prothrombotic environment of the MM patient and can be easily assessed using point-of-care tests should be incorporated into the RAM to increase its sensitivity and optimize its performance. DOACs are expected to replace LMWH, warfarin, and aspirin use. Given the heterogeneity of the MM patient profile and the complex interplay between different factors, we propose that four different levels of VTE risk should be included: no risk, low VTE risk, high risk, and very high risk. 

## Figures and Tables

**Figure 1 cancers-12-00191-f001:**
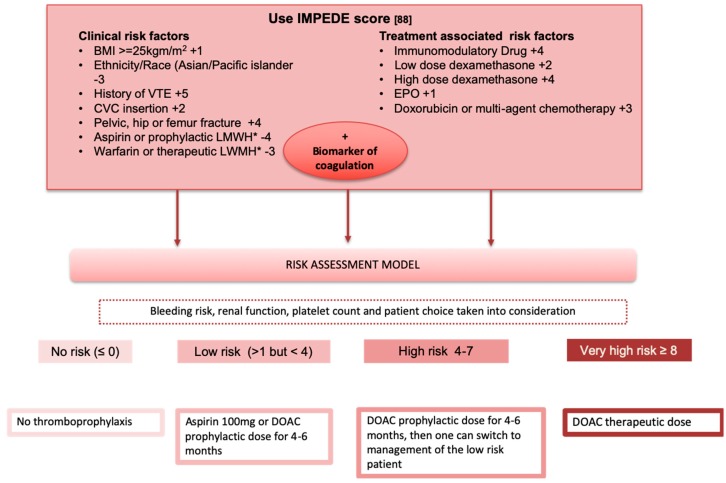
Algorithm for VTE risk assessment and thromboprophylaxis based on Table 1 risk factors; BMI: body mass index, CVC: central venous catheter, LMWH: low molecular weight heparin, EPO: erythropoietin, DOAC: direct oral anticoagulants. * Aspirin, LMWH, or warfarin for other clinical indication prior to treatment initiation for MM.

**Table 1 cancers-12-00191-t001:** International Myeloma Working Group, European Myeloma Network, and National Comprehensive Cancer Network risk stratification algorithm and choice of thromboprophylaxis in patients with multiple myeloma. IMiD: immunomodulatory agent, MM: multiple myeloma, VTE: venous thromboembolism.

Algorithm for MM Patient Risk Stratification		
Patient-Related Risk Factors *ASSIGN 1 Point for Each of the below:*	Disease-Related Risk Factors: *Assign 1 Point for Each of the below:*	Treatment-Related Risk Factors: *Assign Points as Seen below:*
Body mass index >25, Age >75, Personal or family history of VTE, Central venous catheter, Acute infection or Hospitalization, Blood clotting disorders or Thrombophilia, Immobility with performance status of >1, Comorbidities (liver, renal impairment, chronic obstructive pulmonary disorder, diabetes mellitus, chronic inflammatory bowel disease), Race (Caucasian is a risk factor)	Diagnosis of multiple myeloma Evidence of hyperviscosity	IMiD in combination with low-dose dexamethasone (<480 mg/month) *(1 point)* IMiD plus high-dose dexamethasone (>480 mg/month) or doxorubicin or multiagent chemotherapy (*2 points)* IMiD alone (*1 point)* Erythropoietin use (*1 point)*
**Risk stratification and recommended thromboprophylaxis:** 0 points: Low risk ***None*** 1 point: Intermediate risk ***Aspirin at 100 mg*** >1 points: High risk ***Low molecular weight heparin at prophylactic dose or therapeutic dose of warfarin***		

**Table 2 cancers-12-00191-t002:** Risk factors associated with venous thromboembolism in multiple myeloma and studies that have reported the relevant association. CVC: central venous catheter, IMiD: immunomodulatory agent, CABG: coronary artery bypass graft, COPD: chronic obstructive pulmonary disease, NDMM: newly diagnosed multiple myeloma patients, DVT: deep vein thrombosis, NFκB1: nuclear factor kappa B subunit 1.

Patient Related Risk Factors					
**Age**	Brown et al., 2016 [26] hazard of thrombosis for the 35–64 and 65–74 age groups compared to the 18–34 reference group, HR 2.8 for the 75 + age group (1.6–4.8 95% CI)		Baker et al. 2018 [22] Age not identified as risk factor for VTE (*p* = 0.56)	Bagratuni et al. 2013 [27] n = 200, VTEs were more frequent in patients >65 years (8.1% vs. 1.6%)	
**Body mass index ≥30 kg/m^2^** **Family history** **Race**	No specific studies in MM for these risk factors				
**Personal history**	Anaissie et al. 2012 [28] history of VTE was a strong predictor of VTE on univariate analysis (*p* < 0.000005) n = 604				
**Cardiac disease (e.g., symptomatic coronary artery disease, congestive heart failure, or history of stent placement/CABG)**	Brown et al. [26] congestive cardiac failure associated with hazard HR = 1.7 (95% CI, 1.4–2.1), hypertension associated with hazard (HR = 1.2 (95% CI, 1.0–1.3))				
**Other comorbidity**:	**Diabetes mellitus, renal impairment, liver impairment, chronic inflammatory disease, COPD, immobilization, autoimmune disease, recent trauma or surgery, hospitalization, immobility, inherited thrombophilia, use of hormone replacement, acute infection** ***No specific data on these risk factors in patients with MM available***				
**Use of erythropoietin (EPO)**	Anaissie et al. 2012 [28] n = 604 prophylactic EPO (*p* = 0.002; OR, 2.488; 95% CI, 1.432–4.324)	Chalayer et al. 2018 [29] OR 0.49 (95% CI 0.18–3.83)	Knight et al. 2015 [30] n plus lenalidomide: OR 3.21 (1.72–6.01 95% CI, *p* < 0.001)	Galli et al. 2004 [31] n = 199, 8.1% prevalence with EPO vs. 9.3% without, *p* > 0.5)	Leleu et al. 2013 [5] Relative RIsk of VTE 3.46 (0.45–3.7 95% CI, *p* = 0.04)
**Central venous catheter or pacemaker**	Cortelezzi et al. 2005 [32] 12% VTE events in 416 patients with hematologic malignancies and CVC insertion (MM diagnosis seen in 18.8% of pts)				
**Disease-specific risk factors**					
**New diagnosis of MM**	Zangari et al. 2003 [33] (n = 535) newly diagnosed disease (OR, 2.5; *p* = 0.001)				
**Chromosome 11 abnormalities**	Zangari et al. 2003 [33] (n = 535) (OR, 1.8; *p* = 0.048)				
**Microparticle (MP)-associated tissue factor and tissue factor (TF)**	Auwerda et al. 2011 [34]: (n = 122) NDMM; MP-TF levels prior to treatment initiation did not predict VTE, but MP-TF remained elevated in patients who developed VTE 15.1 [10.3–25.2], in contrast to patients not developing VTE (11.4 [7.0–25.2], *p* < 0.001				
**Thrombin lag phase(s)**	Undas et al. 2015 [35] 60 [52–60.5] vs. 50 [36,37,38,39,40,41,42,43,44,45], *p* = 0.01 in patients with VTE				
**Thrombin peak concentration (nmol/L)**	Undas et al. 2015 [35] higher peak concentration associated with VTE; 503.5 (418–550) vs. 344.8 (269–411) in patients without VTE, *p* < 0.001	Leiba et al. 2017 [45] higher peak height values (620 vs. 400 nM, *p* < 0.001) associated with higher VTE risk	Chalayer et al. 2018 [29] 186 nmol/L for patient with VTE vs. 149 nmol/L for not VTE, *p* = 0.22 in univariate analysis	Ay et al. 2011 [25] associated with VTE risk	
**Thrombin peak time (min)**	Chalayer et al. 2018 [29] at baseline; 10.8 min for patients with VTE vs. 9 min for no VTE, *p* = 0.82 in univariate analysis, no significant association with VTE			Ay et al. 2011 [25] associated with VTE risk	
**Endogenous thrombin potential (ETP) (Mxmin)**	Dargaud et al. 2019; ETP higher in MM patients versus controls [46]		Ay et al. 2011 [25] not associated with VTE risk	Leiba et al. 2017 [45] higher EPT (2896 vs. 2028 nMxmin, *p* < 0.001) associated with higher VTE risk	Chalayer et al. 2018 [29] increase in ETP between baseline and cycle 4—no association with VTE
**Thrombin-activatable fibrinolysis inhibitor (TAFI) (mg/mL)**	Undas et al. 2015 [35] higher levels associated with VTE 45.3 (44.6–47.4) vs. 38.9 (33.5–42.3) <0.001				
**Plasminogen activator inhibitory (PAI-1) (IU/mL)**	Undas et al. 2015; [35] higher PAI-1 levels associated with VTE risk 11 (9.9–12.8) vs. 8.3 (6.4–10.5), *p* = 0.004				
**Lower clot permeability and clot lysis**	Undas et al. 2015; [35] in patients with lower clot permeability Ks (10^−9^ cm^2^) and lower D-D_rate_, (maximum rate of increase in D-dimer levels in the lysis assay) associated with higher VTE risk				
**Acquired activated protein C resistance (aAPC-R)**	Zangari et al. 2002 [47] higher proportion of patients with APC resistance developed DVT (5/14 versus 7/38; *p* = 0.04)–41.7% prevalence of APC-R in the group of NDMM who developed VTE		Cini et al. 2010 [48] no difference in VTE occurrence between patients with APCR (6.7% vs. 10.3%, *p* = 1.0)		Elice et al. 2006 [49] higher incidence of VTE with aAPC-R; 1178 patients; 31% versus 12%; *p* < 0.001)
**NFκB1 gene single nucleotide polymorphism**	Bagratuni et al. 2013 [27] NFκB1 and VTE risk: OR 3.76, 95%CI 1–16, *p* = 0.051				
**Factor v. Leiden (R506Q) or G20210A prothrombin mutation**	Cini et al. [48] patients with polymorphisms had not increased VTE rate (10% vs. 9.4%, *p* = 0.27)			Bagratuni et al. 2013 [27] FVLeiden and FIIG20210A not associated with higher VTE rates	
**P-selectin (ng/mL)**	Ay et al. 2008 [50] Elevated P-selectin (>53.1 ng/mL) risk factor for VTE (HR = 2.6, 95% CI, 1.4–4.9, *p* = 003)				
**vonWillenbrand (VWF) increased levels**	Minnema et al. 2003 [51] N = 19 patients on thalidomide VWF-Ag in patients with VTE was 375 ± 121% vs. 235 ± 116% in patients without VTE (*p* = 0.03)			Van Marion et al. 2008 [52] higher levels of VWF not associated with VTE OR 2.69 95% CI 0.71–10.26, *p* = 0.147	
**FVIII (factor VIII)**	Minnema et al. 2003 [51] N = 19 patients on thalidomide FVIII:C was 352 ± 67% vs. 283 ± 114% in patients without VTE (*p* = 0.17)		Cini et al. 2010: [48] elevated FVIII activity not associated with higher VTE rate (10% vs. 7.4% *p* = 0.76)		Van Marion et al. 2008 [52] higher levels of FVIII not associated with VTE occurrence
**Other biomarkers**	Increased D-dimer levels, prothrombin 1 and 2 increased levels, hyperviscosity, antiphospholipid antibodies, lupus anticoagulant—resistance to protein C pathway *No data on these biomarkers and VTE risk*				
**Myeloma Therapy Related** [53]					
**IMiD in combination with:** High-dose dexamethasone (>480 mg/month) Multi-agent chemotherapy Doxorubicin **IMiD alone**					

**Table 3 cancers-12-00191-t003:** Clinical risk assessment models for VTE prediction in MM patients. RAM: risk assessment model; VTE: venous thromboembolism; MM: multiple myeloma; BMI: body mass index; CVC: central venous catheter; LMWH: low molecular weight heparin.

CLINICAL RAMs for VTE in MM	
IMPEDE VTE Score	SAVED Score*
Immunomodulatory drug (+4) BMI ≥ 25 kg/m^2^ (+1) Pathologic fracture pelvis/femur (+4) Erythropoiesis-stimulating agent (+1) Dexamethasone (High-dose) (+4) Dexamethasone Low-Dose (+2) Doxorubicin (+3) Ethnicity/Race = Asian (−3) VTE history (+5) Tunneled line/CVC (+2) Existing use of therapeutic warfarin or low molecular weight heparin (LWMH) (−5) Existing use of prophylactic LMWH or aspirin (−3)	Surgery (within last 90 days) (+2) Asian Race (−3) VTE history (+3) Eight (age >=80 years) (+1) Dexamethasone dose Standard (+1) High (+2) * for patients on IMiD-based regimens only
Stratified risk groups based on weighted scoring system	
Low risk (score ≤3) Intermediate-risk (score of 4–7) High risk (≥8 score)	High risk (score (≥2) Low risk (≤1)
**Missing: recommendation on thromboprophylaxis based on risk groups**	

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
