# Peer review of "Multiple Myeloma and Thrombosis: Prophylaxis and Risk Prediction Tools"

_cancers, 2020, doi:10.3390/cancers12010191_

Round 1

Reviewer 1 Report

This reviews addresses an important issue on the management of MM patients and proposes a useful prophylaxis and risk prediction tool. It is therefore a worthwhile review, however, there are a few minor points for improvement. 

Table 2 in its current format is not particularly well presented or easy to navigate. Perhaps a more cohesive method to collate and present the data in this table could be found?

References within the text are not in a consistent format, sometimes they are in bold, sometimes not, they are presented in an ad hoc manner rather than in numerical order at the end of a sentence and they are unnecessarily listed in individual brackets. For example in line 161 they are listed as  [70] [71, 72] rather than [70-72].

When an abstract that has been presented at ASH is referenced it should be given its full journal listing with journal title, year, volume and article title rather than just listed as 'proceedings of ASH'.

Author Response

Thank you very much for the comments see below for the answers.

Table 2 in its current format is not particularly well presented or easy to navigate. Perhaps a more cohesive method to collate and present the data in this table could be found?

The table has been revised to make the data presentation more cohesive.

References within the text are not in a consistent format, sometimes they are in bold, sometimes not, they are presented in an ad hoc manner rather than in numerical order at the end of a sentence and they are unnecessarily listed in individual brackets. For example in line 161 they are listed as  [70] [71, 72] rather than [70-72]. Have all been made consistent When an abstract that has been presented at ASH is referenced it should be given its full journal listing with journal title, year, volume and article title rather than just listed as 'proceedings of ASH'. Has been corrected

Reviewer 2 Report

I reviewed the manuscript and find it well balanced, comprehensive and of good quality. Based on the review of the literature, they are proposing skeleton for a new algorithm to access no risk/low risk/high and very high risk of thrombosis. I urge them to suggest/define a score for the risk factors and for the proposed categories. 

Author Response

I reviewed the manuscript and find it well balanced, comprehensive and of good quality. Based on the review of the literature, they are proposing skeleton for a new algorithm to access no risk/low risk/high and very high risk of thrombosis. I urge them to suggest/define a score for the risk factors and for the proposed categories.

Answer: Thank you very much for the comment. The algorithm has been updated to include the IMPEDE Risk score for weighting of risk factors and determination of the risk categories

Reviewer 3 Report

Overall this is an important review article which addresses 1) Pathophysiology of thrombosis in MM 2) Prophylaxis strategies in MM 3) Risk prediction scores. The authors have summarized a wealth of important and pertinent information. My comments are as below:

Abstract (line 21):

"eventually lead to the elimination of this significant complication". I think we will likely decrease rates of VTE, but i found the word elimination too strong. 

Introduction

"Thrombotic events do, however, have an adverse impact as they lead to treatment interruption". MAY lead to treatment interruption.  " Data from other cancer patients demonstrate as expected increased costs associated with the long term use of 39 pharmaceutical agents for the treatment of thrombosis, the need for hospitalization, and increased  risk of complications as well as adverse effects on patient's quality of life". Reference missing "In a recent report, the rate of compliance with guidelines was only 66% in a cohort of patients who received lenalidomide based regimens." Reference missing

2. Understanding the complex procoagulant profile of MM patients

Up to date: ? To date

3. Risk assessment tools

Do the authors have any comments on why the Khorana risk sore for cancer patients cannot be extended to MM? Also, why do the authors think the risk factors for each of the RAM (IMPEDE and SAVED) score are so different when they are derived and validated in similar settings?

Conclusion and Recommendations

How will relapsed, refractory patients be different from newly-diagnosed for RAM? Up to date again used incorrectly "Low cost and simple assessment tools that do not require high level expertise are pre-requisites for the selection of a suitable biomarker". In addition to the sensitivity and specificity of the marker "could perhaps signify the fact that no single marker will ever be able to capture it" Its possible a single marker may be able to capture at some later point, maybe authors should instead emphasize the complexity and likely the interactions of multiple factors Another disadvantage of oral adminstration route is polypharmacy which is especially difficult in older patients with myeloma who form the bulk of all patients.

Tables

Table II: the bolding and non-bolding at times does not make sense, what is the time to peak at baseline (thrombin peak time??). Myeloma therapy related factors have period prior to sentence. 

Figure 1

Why is the biomarker of coagulation in a separate bubble? If the outcome of it includes no thromboproph, ASA or DOAC, why are those factors also included under the clinical risk factors?

Author Response

Thank you very much for the comments, see below for the answers.

Overall this is an important review article that addresses 1) Pathophysiology of thrombosis in MM 2) Prophylaxis strategies in MM 3) Risk prediction scores. The authors have summarized a wealth of important and pertinent information. My comments are as below:

Abstract (line 21): 

"eventually lead to the elimination of this significant complication". I think we will likely decrease rates of VTE, but i found the word elimination too strong. Has been changed to reduction

Introduction

"Thrombotic events do, however, have an adverse impact as they lead to treatment interruption". MAY lead to treatment interruption. Done

 " Data from other cancer patients demonstrate as expected increased costs associated with the long term use of 39 pharmaceutical agents for the treatment of thrombosis, the need for hospitalization, and increased  risk of complications as well as adverse effects on patient's quality of life". Reference missing "In a recent report, the rate of compliance with guidelines was only 66% in a cohort of patients who received lenalidomide based regimens." Reference missing Thank you, has been added

Understanding the complex procoagulant profile of MM patients. Up to date: ? To date has been corrected Risk assessment tools

Do the authors have any comments on why the Khorana risk sore for cancer patients cannot be extended to MM? Also, why do the authors think the risk factors for each of the RAM (IMPEDE and SAVED) score are so different when they are derived and validated in similar settings?

The following has been added in line 260.” Despite the fact that the two scores were developed and validated in similar settings there are significant differences. One reason could be linked to the fact that the SAVED score was developed selecting only MM patients receiving IMiDs. The methodological approach followed is also not identical. Finally each score possibly capture VTE risk in a unique manner which has however significant overlap with the other score given the particularly multifaceted nature of thrombosis in MM patients.”

Thank you for the comment:

The following has been added in line 208 to demonstrate that the Khorana score does not perform well in MM patients most likely due to the great impact of treatment related parameters in the thrombotic risk.

The Khorana score cannot be extended to MM patients as it does not accurately predict VTE in the MM population. [87] A RAM specific for MM which includes treatment related parameters to adequate reflect thrombotic risk is required.”

Conclusion and Recommendations

How will relapsed, refractory patients be different from newly-diagnosed for RAM? A possible suggestion would be that a new diagnosis should be incorporated as an additional risk factor in risk assessment process. The following sentence has been added: “Currently no risk assessment tool makes a distinction between NDMM and RRMM patients. A new MM should perhaps in the future be included in RAMs as an additional risk factor.” (line 346)

Up to date again used incorrectly "Low cost and simple assessment tools that do not require high level expertise are pre-requisites for the selection of a suitable biomarker". Corrected

In addition to the sensitivity and specificity of the marker "could perhaps signify the fact that no single marker will ever be able to capture it" Its possible a single marker may be able to capture at some later point, maybe authors should instead emphasize the complexity and likely the interactions of multiple factors.

The sentence has been changed to “The inability to identify so far a generic biomarker to accurately reflect the above processes is perhaps a reflection of the complex and heterogeneous coagulation profile of MM patients which results from the interaction of multiple factors. “

Another disadvantage of oral adminstration route is polypharmacy which is especially difficult in older patients with myeloma who form the bulk of all patients.

This comment has been added at line 300 “An additional issue associated with the oral route of administration is however polypharmacy which is very relevant in these patients.”

Tables

Table II: the bolding and non-bolding at times does not make sense, what is the time to peak at baseline (thrombin peak time??).

Bolding and non-bolding has been revised to make more sense. Time to peak has been changed to thrombin peak time.

Myeloma therapy related factors have period prior to sentence. Has been corrected

Figure 1

Why is the biomarker of coagulation in a separate bubble?

The bubble to separate the biomarker is supposed to make the point that no such biomarker has been identified yet. Have changed the figure to include the bubble within the box rather than outside it.

If the outcome of it includes no thromboproph, ASA or DOAC, why are those factors also included under the clinical risk factors? It’s meant to be ASA or DOAC for another clinical indication prior to the assessment of thromboprophylaxis for MM treatment. An asterisk has been placed with an explanation to make this clear.

Reviewer 4 Report

I read the manuscript with great interest. The authors described the updated information regarding the risks and prediction tools of thrombosis and the prophylaxis in multiple myeloma. The manuscript is well written and I do believe that it will be helpful and useful for daily practice. The manuscript is acceptable for publication if minor points as describe below is corrected.

Check again carefully and rewrite in order the reference number. (Ex. In lines 34 in page 1, [5][6][7][3] to [3, 6-7]) Correct misspelled abbreviation in table 2 of page 4. (vonWillebrand factor:"WVF" to "VWF") In lines 222 in page 8, eight years old? (eighty years old is correct?) In lines 258 in page 9, "Its is use is" is not correct. Correct it. In lines 261 in page 9, "NCNN" is "NCCN"? In lines 303 in page 10, "75-80.000mm3" is not correct. Correct it.

Author Response

I read the manuscript with great interest. The authors described the updated information regarding the risks and prediction tools of thrombosis and the prophylaxis in multiple myeloma. The manuscript is well written and I do believe that it will be helpful and useful for daily practice. The manuscript is acceptable for publication if minor points as describe below is corrected.

Thank you very much for the comments

Check again carefully and rewrite in order the reference number. (Ex. In lines 34 in page 1, [5][6][7][3] to [3, 6-7]) this has been corrected for all references.

Correct misspelled abbreviation in table 2 of page 4. (vonWillebrand factor:"WVF" to "VWF") has been done

In lines 222 in page 8, eight years old? (eighty years old is correct?) Has been corrected to eighty

In lines 258 in page 9, "Its is use is" is not correct. Correct it. Has been corrected

In lines 261 in page 9, "NCNN" is "NCCN"? done

In lines 303 in page 10, "75-80.000mm3" is not correct. Correct it. Done thank you
